# What really matters for global intergenerational mobility?

**Khanh Duong**  *

Department of Sociology, Maynooth University, Kildare, Ireland

* duongkhanhk29@gmail.com

**Data Availability Statement:** In line with the principles of Open Data and Research, the data and codes pertaining to this research have been made publicly available. You can access them at the following locations: - Research data: https://doi.org/10.6084/m9.figshare.24916074.v2 - Research

## Abstract

This study investigates the genuine impacts of education expansion, education inequality, and parental dependency on intergenerational mobility. It utilizes data from the Global Database on Intergenerational Mobility for 153 countries and cohorts born between the 1940s and 1980s. By employing a causal machine learning approach to address confounding problems, this research reveals that education expansion can promote intergenerational mobility to a certain extent. However, its effectiveness is partially diminished by education inequality and may be ineffective if parental dependency exists at a high level. Furthermore, this study also indicates that while gender inequality in intergenerational mobility still exists, its degree has been significantly reduced across generations. When compared to parental dependency, gender effects are far less important. Therefore, there is a need to reassess the roles of parental dependency and gender bias in intergenerational mobility, especially when parental dependency is currently underestimated, and gender bias is overemphasized.

## 1. Introduction

The issue of increasing global inequality has emerged as a pressing concern that demands attention and action in the modern world. One lens through which this inequality is understood is the Great Gatsby Curve [1]. This concept highlights the relationship between income inequality and intergenerational mobility, revealing that societies with higher levels of income inequality tend to experience lower rates of intergenerational mobility. A closer look at specific countries and regions further illustrates this association. For instance, Scandinavian countries like Denmark, which exhibit relatively low levels of inequality, have been found to have higher levels of intergenerational mobility compared to liberal welfare-state countries such as the United States, which have higher levels of inequality [2]. Research finds that the level of intergenerational immobility, which refers to the persistence of socio-economic status across generations, varies depending on the welfare regime in place. Notably, compared to liberal and social-democratic welfare regimes, under a conservative welfare regime, the level of intergenerational immobility tends to be the highest. This regime effectively perpetuates the influence of parental backgrounds on the wealth and socio-economic status of their children [3]. The conservative welfare regime, characterized by its emphasis on traditional values and limited

codes: https://doi.org/10.6084/m9.figshare.
24916068.v1.

**Funding:** The research conducted in this
publication was funded by the Irish Research
Council under award number GOIPG/2021/441.

**Competing interests:** The authors have declared
that no competing interests exist.

state intervention, tends to reinforce existing social hierarchies and inequalities, thereby hindering upward mobility for individuals from disadvantaged backgrounds [4].

Not only is upward mobility influenced by existing social hierarchies and inequalities, but it is also shaped by familial factors that have been formed over generations. This phenomenon is often referred to as intergenerational persistence, whereby the educational advantages or disadvantages inherited from parents significantly shape an individual's educational trajectory and subsequent socio-economic status. Indeed, extensive research has consistently highlighted the strong correlation between children's educational outcomes and their parents' educational attainment. Studies have shown that children from families with higher levels of educational attainment are more likely to achieve better educational outcomes themselves [5, 6]. Specifically, in Germany and the EU in general, Jan Skopek and Giampiero Passaretta [7, 8] conducted research on socioeconomic inequality from infancy to adolescence. They found that the socioeconomic gaps were formed and expanded during the pre-schooling phase, and schooling decreases these inequalities. This implies that familial conditions are key to determining a person's socio-economic status, thereby suggesting that parental dependency is one of the factors hindering upward mobility–often referred to as 'like father, like son, like mother, like daughter' [9].

To promote intergenerational mobility, education has been identified as a key factor [10]. Research suggests that countries with higher levels of intergenerational educational mobility, indicative of more equal access to education across generations, are likely to experience both economic growth and social progress [11]. Thus, education emerges as a crucial driver in breaking the cycle of inequality and fostering upward mobility across generations. However, educational investment policies do not always resolve this issue [12]. Deirdre Bloome et al. [13] found that the expansion of higher education enabled upward mobility for low-income individuals who completed college, reducing persistence to some extent in the US. However, this reduction was insufficient to offset the overall increase caused by growing educational inequality and rising educational returns. Another factor that prevented a further increase in intergenerational persistence was the decreasing dependence of adult income on parental income within educational groups. Additionally, evidence from the UK also highlights the existence of educational inequality, which hinders the positive impact of educational expansion on social mobility [9]. Alongside the obstruction of educational inequality and parental dependency, the effectiveness of educational expansion itself also comes into question. Educational expansion may lead to 'over-education' and amplify the influence of parents' social connections on their children's education, potentially resulting in horizontal inequity and inefficiencies in human capital accumulation. Furthermore, it may contribute to more persistent intergenerational immobility, particularly in higher education [14].

*Q1. How much is the true effect of education expansion on intergenerational mobility, under the hindrance of social inequality and parental dependency?*

I have discussed the impact of educational expansion on intergenerational mobility in relation to social inequality and parental dependency, but the link between social inequality and parental dependency has not been discussed above. Continuing from the previous section, I consider a type of inequality that exists not only in society but also within a family–that is gender inequality, as a key to explaining the general relationship existing in social inequality and parental dependency, when income or race inequality only exists in society, but is levelled within a family. According to Gender theory [15], biological factors of gender do not directly impact intergenerational mobility but through social norms and cultures. In research on the impact on intergenerational mobility, the gender effect through social inequality typically manifests through differential treatment in terms of equal opportunities for education and

employment between men and women [16, 17]. Many studies have investigated this dimension of the gender effect, notably Philipp Bach et al. [18] in the US on the gender wage gap and Grace Chisamya et al. [19] in Bangladesh and Malawi on gender inequities in schools and communities. On the other hand, the gender effect through parental dependency manifests through how parents treat their sons and daughters within the family. For example, in Mexico, parents are more likely to transfer socioeconomic resources to their married sons than married daughters [20]. In India and China [21, 22], gender biases and discrimination, such as underestimation of daughters' abilities and lower expectations, affect education mobility.

In addition to social inequality and parental dependency, gender inequality in intergenerational mobility has garnered significant attention from the academic community. Evidence of gender inequality is apparent in numerous countries worldwide, such as Greece [23], Brazil [24], Spain [25], and India [22], as well as in developed regions like the EU [26] and developing regions like Sub-Saharan Africa [27]. Beyond the gender effects via social inequality and parental dependency, gender effects can also change according to spatial and temporal factors (contextual factors). Research has shown that gender differences can vary across places. For instance, in India, a persistent gender gap in educational mobility exists in rural and less-developed areas, while women in urban and developed regions, particularly from lower castes, have experienced significant improvements [22]. Similarly, in Turkey, educational outcomes for daughters are less dependent on their parents' educational achievements in more developed regions, but no comparable relationship is found for males [11]. Research also indicates that gender differences have reduced over time. Julie Park et al. [28] documented that the second generation of post-1965 immigrant women have made significant socioeconomic advances over the last generation. Olivetti & Paserman [29] found that both father-son and father-daughter elasticities were flat during the nineteenth century, increased sharply between 1900 and 1920, and declined slightly thereafter. According to a World Bank report, girls in high-income economies have surpassed boys in terms of absolute intergenerational mobility and tertiary education rates [30]. This gender gap reversal began with the 1960s cohort and has since grown. Similar trends are observed in developing economies, with women rapidly closing the gender gap in absolute mobility and matching men in tertiary education [30]. Recently in 2021, Alberto Alesina et al. [31] conducted a meta-data analysis in Africa and found no evidence of systematic gender gaps in intergenerational mobility and argued that geographic and historical factors play a significant role in intergenerational mobility in this region. These findings, and the evidence of reduced gender inequality in recent generations, have raised a new research question about the extension of gender effects in this current time.

*Q2) Is intergenerational mobility still seriously gender-biased in the multidimensional analysis with social inequality, parental dependency and contextual factors?*

This study aims to measure the significance of factors such as education expansion, social inequality, parental dependency, and gender effects on intergenerational mobility. As depicted in Fig 1, social inequality and parental dependency have long been recognized as major barriers to upward mobility (denoted as 1 and 2), and they also undermine efforts in education expansion to promote upward mobility (denoted as 3). Furthermore, the question regarding the effectiveness of education expansion itself was raised by Hai Zhong [14] in the context where over-education might not facilitate an individual's upward mobility. While education expansion, social inequality, and parental dependency have a direct impact on upward mobility, gender effects fundamentally influence upward mobility through indirect mechanisms in the family, society, and context. As gender differences are gradually eliminated across generations, and as Alberto Alesina et al. [31] no longer found strong evidence of gender inequality in Africa in recent research, these findings necessitate a revisit of the extent of gender

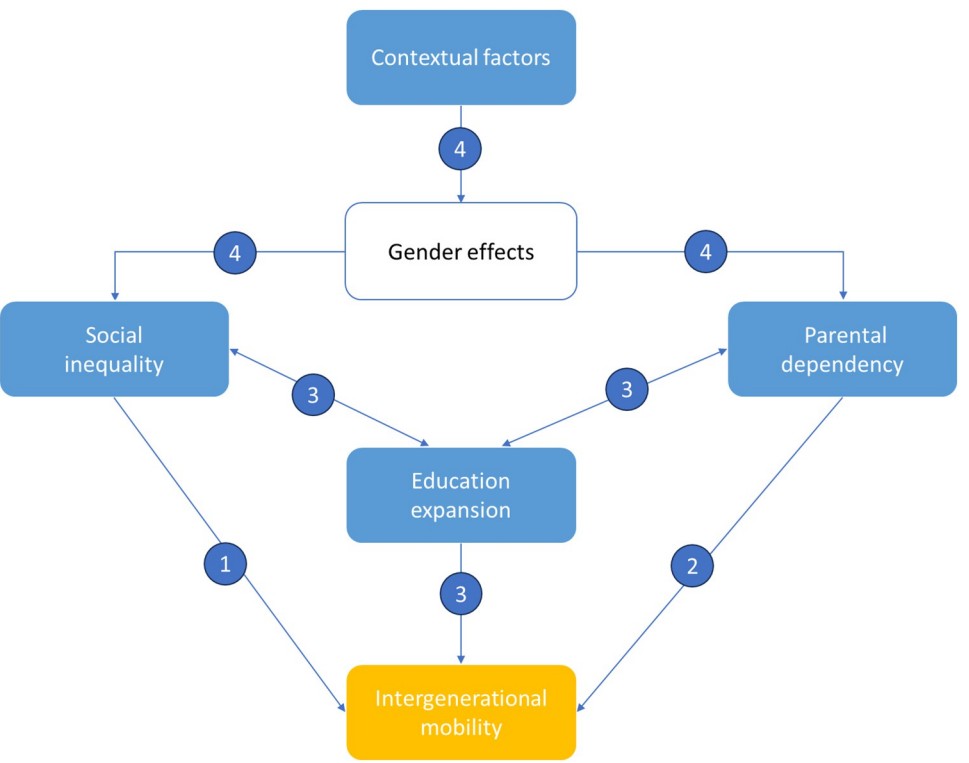

**Fig 1. The analytical framework.** This figure is a compilation of past findings by the author.

inequality in intergenerational mobility, especially when its multidimensionality and dynamics challenge traditional methods. Utilizing causal machine learning recently developed by Chernozhukov et al. [32] to address confounding problems, I re-examine the true effect of education expansion, social inequality, parental dependency, and gender effects in complex multidimensional relationships. The originality of this research lies in demonstrating that promoting social mobility primarily involves lowering the level of parental dependency, rather than addressing social inequalities (such as education inequality or gender bias)—as many studies are concerned with—or education expansion—as governments are implementing. In the following section, I describe the data and analytical methods used in this study, then present the research findings and discuss them in relation to the current literature.

## 2. Materials and methods

### 2.1 Research materials

**2.1.1 Data sources.** The data source for this study is the Global Database on Intergenerational Mobility (GDIM), which provides information on cohorts born in the 1940s, 1950s, 1960s, 1970s, and 1980s [33]. In the GDIM, each observation represents a distinct combination of code- cohort-parent-child. The variable code denotes the country code. The variable cohort designates the generation born within a specified decade. The variable parent refers to the parents of individuals in this cohort and is disaggregated by the type of parental educational attainment (Mothers/Fathers/Average/Max). The variable child pertains to the educational attainment of the child and is disaggregated by the type of child's educational attainment (Sons/Daughters/All). Hence, the GDIM has 12 estimates by each country and cohort (by type of parent and by type of child) or 12 units of analysis for each survey (uniquely identified by

code-cohort combinations). The study includes a total of 153 countries classified according to their economic development, fragility, and region (see S1 Appendix for data summary). Of the 153 countries, 115 are considered developing economies, and they are distributed across various regions such as South Asia, Sub-Saharan Africa, Latin America and the Caribbean, Europe and Central Asia, and East Asia and the Pacific. Fragility classification (or World Bank Fragile and Conflict-affected Situations) is used to indicate the level of vulnerability and instability of a country, and it is assessed based on factors such as conflict, political instability, and weak governance. The study also provides information on the region in which each country is located, such as the Middle East and North Africa, Sub-Saharan Africa, Europe and Central Asia, South Asia, and East Asia and the Pacific. This comprehensive classification of countries allows for a nuanced analysis of intergenerational mobility across different types of countries and regions, providing valuable insights into the factors that affect social mobility in different contexts.

**2.1.2 Variable definitions.** *Upward mobility*, the main concept of this study, refers to the degree of absolute mobility in the GDIM. This indicator shows the probability that a child achieves a higher level of education than their parent. Education is a suitable proxy for upward mobility because people value equal opportunities (e.g., access to education) more than equal outcomes (e.g., income or wealth) [34]. Additionally, income is a problematic variable to compare across countries and generations due to different standards of living and inflation. Education, by contrast, is a more stable and comparable measure, and a crucial predictor of income [35].

*Education expansion*, as referenced in the study of Deirdre Bloome et al. [13], represents the difference between the mean years of schooling of children and the mean years of schooling of parents. It signifies the magnitude of educational expansion within a specific population. Higher values of education expansion indicate a greater increase in educational attainment among younger generations, potentially contributing to upward mobility. It should be noted that the expansion of education can potentially elevate the educational levels of a nation as a whole, but it does not necessarily imply a transformation in the internal social structure of that nation.

*Education inequality*, as referenced in the study by Blanden and Macmillan [36], is quantified by the disparity between the standard deviation of educational years among children and that among parents. This metric encapsulates the variations in educational achievement within a given population. A larger value of education inequality implies a more significant divergence in educational outcomes, which can potentially impact the opportunities for upward mobility among individuals from diverse socio-economic backgrounds. It is worth noting that within the analytical framework in Fig 1, education inequality could conceivably supplant the role of social inequality as education serves as a key indicator for computing intergenerational mobility measures in this study.

*Parental dependency*, as referenced in the study of Deirdre Bloome et al. [13], on the other hand, is captured by the correlation coefficient between children's years of schooling and their parents' years of schooling, as defined by GDIM. The use of the correlation coefficient allows for an accurate measurement of parental dependency, especially in the context of educational expansion which refers to the general trend of improving education in subsequent generations. A higher correlation coefficient, therefore, suggests a stronger association between parental education and children's educational attainment. This is a more accurate interpretation of parental dependency than maintaining the same level of education across generations, which may not be sufficient in the presence of educational expansion.

For *the gender dimension*, this research uses the child variable in GDIM which represents the gender of the child whose intergenerational mobility is being measured. It includes three

categories: 'sons' representing sons' intergenerational mobility, 'daughters' representing daughters' intergenerational mobility, and 'all' representing the intergenerational mobility of both sons and daughters combined. To assess gender inequality, this variable is transformed into a binary value, with 1 assigned if the parent category is 'daughters' and 0 assigned for other categories. To obtain more nuanced results, the parent variable in GDIM is used to categorize the type of parent's education for examining mobility. This variable is transformed into a binary value, with 1 assigned if the parent category is 'mothers' and 0 assigned for other categories.

To account for the indirect impacts of gender inequality via *contextual factors*, I have included three variables in my analysis: 'fragile,' 'developing,' and 'region.' In the context of the GDIM, the variable 'fragile' assesses a country's fragility or instability based on the World Bank's classification of countries deemed fragile or conflict-affected. The variable 'developing' categorizes countries into two groups based on their gross national income per capita: developing economies, and high-income economies. The variable 'region' captures the geographic, political, cultural, and social distinctions among countries. It refers to the seven regions classified by the World Bank, namely East Asia and Pacific, Europe and Central Asia, Latin America and Caribbean, Middle East and North Africa, North America, South Asia, and Sub-Saharan Africa. By incorporating these factor variables, the analysis can capture the multidimensional aspects of gender inequality and its relationship to intergenerational mobility, considering factors such as a country's fragility, economic development, and regional characteristics.

To assess the significance of these contextual variables, I conducted a preliminary analysis by generating violin charts (see Fig 2), which combine a box plot and a kernel density plot to visually represent data distributions and identify data peaks. These charts allow for a comparison of means across different levels of the variables, and an analysis of variance (ANOVA) test was performed. The null hypothesis for the ANOVA test assumes that all population means are equal, while the alternative hypothesis suggests that at least one population mean is different from the others. A p-value below the chosen significance level (typically $p < 0.05$) indicates that there is significant evidence to reject the null hypothesis, indicating differences in mean values among groups. The results of the analysis indicate that there are significant differences in mobility levels between developing countries and high-income economies, with developing countries exhibiting lower mobility. Additionally, fragile countries show significantly lower mobility compared to non-fragile countries. Moreover, the analysis reveals variations in mobility levels across different regions. Based on these findings, it is evident that these variables demonstrate significant differences in relation to intergenerational mobility. As a result, they meet the criteria for inclusion in the subsequent modelling process.

## 2.2 Research methods

**2.2.1 Data pre-processing.** To compare effect sizes, the dataset (see *S2 Appendix*) was standardized using log-transformation. However, some numeric variables in the pre-modelled data contained negative values, making log transformation impossible. To address this issue, an offset was applied to these negative values by taking the absolute value of the minimum value in the entire dataset and adding 1. It should be noted that traditional standardization (Z-score normalization), which scales variables to have zero mean and unit variance, did not work for this study because it created a very high coefficient of variation due to the small mean value of the original data. Categorical variables were dummy encoded. In dummy coding, for N categories in a variable, N-1 binary variables are created. Each binary variable represents one category and takes the value 1 if the observation belongs to that category and 0 otherwise. The category that is not represented by a binary variable is called the reference category. In

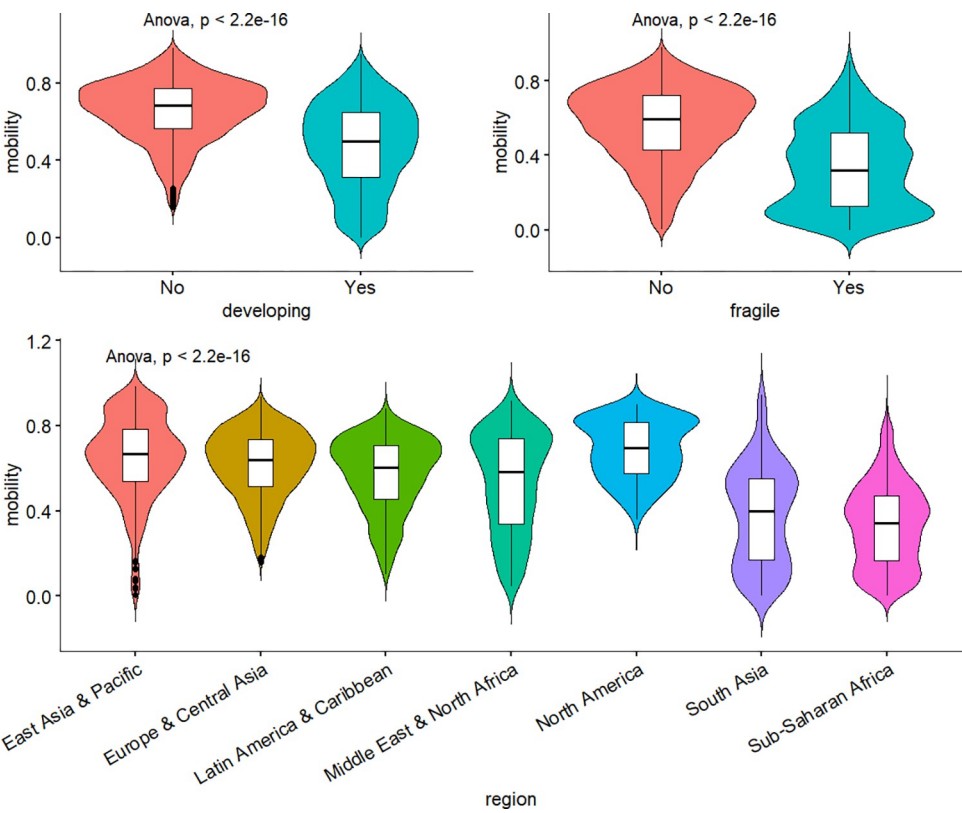

**Fig 2. Upward mobility around the world.** Notes: The data for this estimation comes from the pre-modelled dataset (see S2 Appendix). The method used for comparing means is ANOVA and p is its p-value.

this study, the reference category for 'cohort' (which decade individuals were born in) was 1940 and for 'region' it was East Asia & Pacific. The choice of the reference category was set to default as the first category. Before standardization, 'mobility' was country-centred by subtracting the country mean from each observation within that country. This technique effectively removes fixed effects (or country-specific effects). The final dataset for modelling is summarized in S3 Appendix.

**2.2.2 Empirical strategy.** It is evident that the analysis framework reveals potential confounding issues that may influence the outcome of interest, are related to the investigated factors and may introduce bias, thus making it difficult to determine the true relationship between the variables under consideration. Traditional methods of panel data analysis are inadequate to fully address this problem. To overcome this challenge, I employed Causal Machine Learning methods [37] (see Fig 3 for more details). Consequently, the estimation model with $\theta_0$ being the target effect takes the form:

$$Y = D\theta_0 + l_0(X) + \epsilon, \quad \mathbb{E}[\epsilon|D, X] = 0$$

$$D = m_0(X) + \rho, \quad \mathbb{E}[\rho|X] = 0$$

where $Y$ represents the outcome variable, or 'mobility,' $D$ denotes the treatment variable, or target variable (with estimates for each of the following: education expansion, education inequality, and parental dependency), $X$ signifies the confounding variables which affect both $Y$ and $D$, while $\epsilon$ and $\rho$ signify the error terms. This modelling framework captures the causal

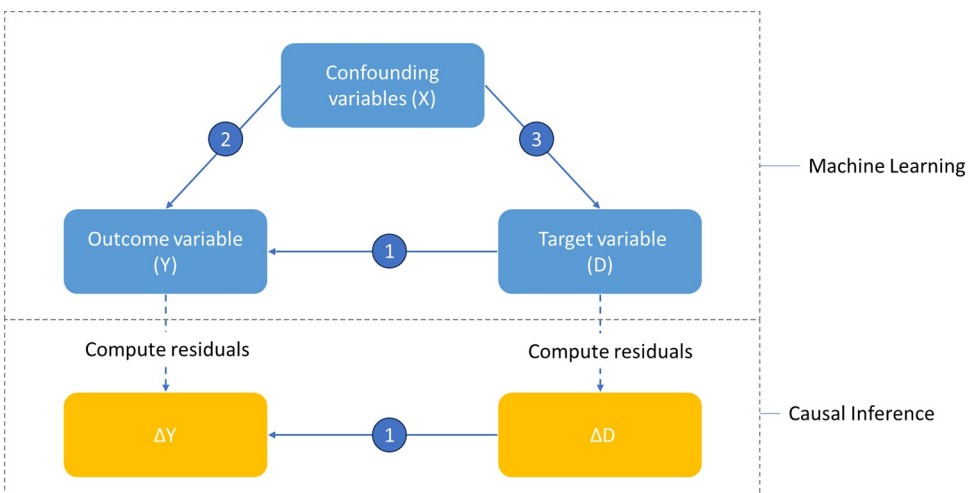

**Fig 3. How do causal machine-learning techniques address confounding problems?.** Notes: In the given figure, the primary effect of interest (denoted as 1) is the impact of D on the outcome variable Y. However, this effect is confounded by variables X that influence both D and Y (denoted as 2 and 3). To address this problem, causal machine learning is utilized, fundamentally consisting of two parts. The machine learning component used to generate D' and Y'–these are the corresponding parts of D and Y that are explained by the confounders X, and the parameters of this process are referred to as the nuisance parameters (such as the indices representing the impact of X on D and Y). The causal inference component will perform estimations from the residuals of Y and D after removing D' and Y'. The orthogonalization technique used in the process ensures that the target effect is not influenced by confounders (in other words, it is invariant to the nuisance parameters). It is also noted that these causal machine-learning techniques only report the target effect.

relationship between $D$, $X$, and $Y$, recognizing that $X$ exerts influence on both $Y$ and $D$. Within the context of this study, it is essential to acknowledge the intricate interconnections among significant factors, such as educational expansion, educational inequality, and parental dependency, as they collectively shape mobility outcomes [38]. When the treatment variable ($D$) is set to represent education inequality, for example, the inclusion of confounding variables ($X$), which consist of variables like educational expansion and parental dependency, leads to a multicollinearity problem. The coefficient estimates of the model can fluctuate significantly based on which other predictor variables are included in the model. The precision of the coefficient estimates is reduced, which makes the p-values unreliable [39]. Thus, the presence of multicollinearity poses challenges in disentangling the individual effects of each factor on intergenerational mobility. To address this estimation predicament, orthogonalization techniques are employed to eliminate the influence of control variables on both the outcome and treatment variables. Specifically, two orthogonalization techniques are utilized: Partialling-out Lasso [40] and Partial Linear Regression (PLR) [32]. The former assumes linear associations X↦Y, D (or $l_0(X) = X\beta_0$ and $m_0(X) = X\pi_0$), with the imposition of Rigorous Lasso effects, while the latter encompasses more complex, unseen relationships X↦Y, D, employing Double Machine Learning (DML). The concurrent implementation of these orthogonalization techniques serves as a robustness check, ensuring the validity and reliability of the analysis.

The orthogonality principle underlying these methods [32, 40] can be described as follows. I seek a score function $\psi(W, \theta, \eta)$, where $W = (Y, D, X)$ and $\eta$ represent the nuisance parameters, in order to satisfy the following conditions: $\mathbb{E}\psi(W, \theta_0, \eta_0) = 0$, $\frac{\partial}{\partial \eta}\mathbb{E}\psi(W, \theta_0, \eta_0) = 0$. Here, $\psi$ ($W; \theta, \eta$) is defined as $(Y - l(X) - \theta(D - m(X)))(D - m(X))$, and $\eta$ is composed of the functions $l$ and $m$, with $\eta_0 = (l_0, m_0)$. Specifically, $l_0(X)$ denotes $\mathbb{E}[Y|X]$, and $m_0(X)$ denotes $\mathbb{E}[D|X]$. The score function $\psi$, with $W = (Y, D, X)$, $\theta_0$ as the parameter of interest, and $\eta$ representing nuisance functions with a population value of $\eta_0$, plays a key role in the inference procedure. It

satisfies the moment condition $\mathbb{E}\psi(W, \theta_0, \eta_0) = 0$, where $\theta_0$ (the target effect) is the unique solution, and it also adheres to the Neyman orthogonality condition $\frac{\partial}{\partial\eta}\mathbb{E}\psi(\omega_i, \alpha_0, \eta_0) = 0$. This condition ensures that the moment condition used for identifying and estimating $\theta_0$ remains unaffected by small perturbations of the nuisance function $\eta$ around $\eta_0$.

Machine learning techniques often lead to overfitting bias, which is where *cross-validation* comes into play [32]. The fundamental principle of cross-validation is to address bias in parameter estimation by dividing the data into folds and estimating nuisance functions and the parameters of interest in separate samples. This approach helps alleviate overfitting and misspecification problems by minimizing the impact of any particular subset of data. The cross-validation for the double machine learning procedure operates as follows. To begin, an assumption is made regarding a full sample $(W_i)_{i=1}^{N}$, where $W = (Y, D, X)$, and a Neyman-orthogonal score function $\psi(W; \theta, \eta)$ is employed. Subsequently, a K-fold random partition $(I_k)_{k=1}^{K}$, obtained by folding the observation indices {1,...,N} is utilized, with each fold $I_k$ having a size of N/K. For every $I_k$, a machine learning estimator of $\eta_0$, denoted as $\hat{\eta}_{0,k}$, is constructed using the out-of-sample data $(W_i)_{i\notin I_k}$. Finally, the estimator for the causal parameter $\tilde{\theta}_0$ is constructed as the solution to the equation:

$$\frac{1}{N}\sum_{k=1}^{K}\sum_{i\in I_k}\psi(W_i; \tilde{\theta}_0, \hat{\eta}_{0,k}) = 0$$

However, the issue of gender effects on mobility in question Q2 cannot be fully resolved by the aforementioned techniques due to the inherent challenge of disentangling the effects of $X$ and $D$ on $Y$, particularly when gender bias factors are intertwined with socioeconomic factors [15]. In a manner akin to the research conducted by Philipp Bach et al. [18] on the multidimensional gender effects with socio-economic characteristics on the gender wage gap in the US, a rigorous methodology is employed in this instance to tackle this problem. Specifically, the Partialling-out Lasso (with interactions) [40] and the Interactive Model Regression (IRM) [32] (as alternatives to the PLR) are utilized. In the context of Partialling-out Lasso with interactions, the treatment variable $D$ encompasses a set of gender variables, including itself and interaction terms with other socioeconomic variables, while $X$ represents the control variables and their corresponding interaction terms. This analytical approach remains consistent. In the meanwhile, in the IRM framework, the term $D\theta_0 + l_0(X)$ is replaced by $g_0(D, X)$, indicating the inseparability of the effects of $D$ and $X$. The estimation of $\theta_0$ is equivalent to determining the average treatment effect (ATE), using a novel score function:

$$\psi(W; \theta, \eta) := (g(1, X) - g(0, X)) + \frac{D(Y - g(1, X))}{m(X)} - \frac{(1-D)(Y - g(0, X))}{1 - m(X)} - \theta$$

Here, $W$ denotes the variables $(Y, D, X)$, while $g(D, X)$ represents $\mathbb{E}[Y|D, X]$, and $m(X)$ signifies $\mathbb{P}[D = 1|X]$. The parameters $\eta$ correspond to $(g, m)$, with $\eta_0 = (g_0, m_0)$. This advanced methodology is employed to address the intricate nature of gender effects and ensure a more robust analysis.

## 3. Results and discussion

### 3.1 How much is the true effect of education expansion on intergenerational mobility?

Prior to conducting the estimation, a preliminary analysis was undertaken to examine the relationship between mobility and three key factors: education expansion, education inequality,

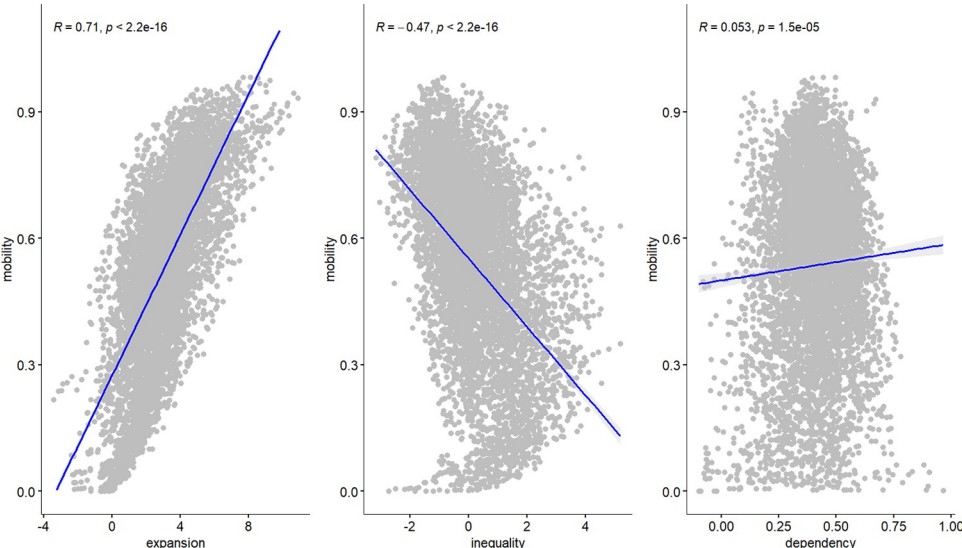

**Fig 4. What really matters for intergenerational mobility?.** Notes: The sample size N = 6,725. The data for this estimation is derived from the pre-modelled dataset (refer to S2 Appendix). R represents Pearson's correlation coefficient, and p denotes its p-value.

and parental dependency. A scatter plot with a blue trend line was employed for this analysis. Pearson's correlation coefficient (R) and its associated p-value were used to quantify the strength and significance of these relationships. Fig 4 presents the findings, revealing a distinct positive relationship between education expansion and mobility, indicated by an R-value of 0.71 ($p < 0.01$). Conversely, a noticeable negative relationship between education inequality and mobility was observed, with an R-value of -0.47 ($p < 0.01$). However, the relationship between parental dependency and mobility appeared less pronounced, displaying a weak positive association with an R-value of 0.053 ($p < 0.01$). Following the preliminary analysis, the data were pre-processed, and Partialling-out Lasso and Double ML estimation techniques were applied. The results of these estimations are visually represented in Fig 5.

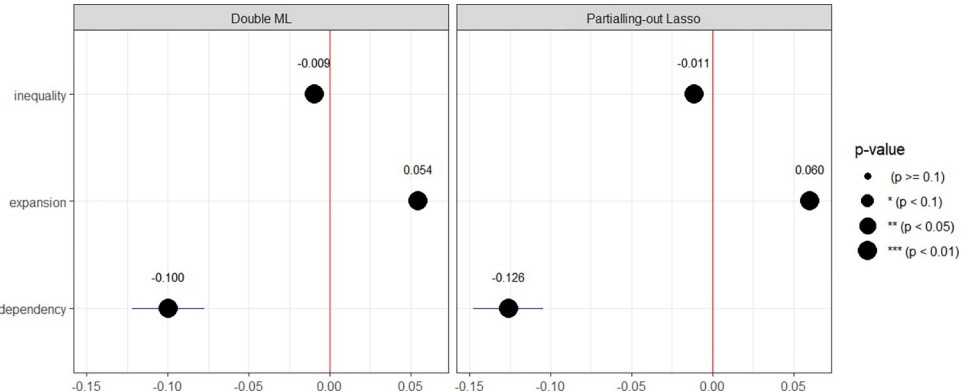

**Fig 5. Does education expansion work for promoting upward mobility?.** Notes: This Fig refers to the table format provided in S4 Appendix. The sample size N = 6,725. In this study, the outcome variable is upward mobility. For each estimate, the treatment variables (target variables) are either (education) inequality, (education) expansion, or (parental) dependency. The confounding variables consist of all other variables present in the dataset, which are not directly considered treatment or outcome variables.

The estimation results of Partialling-out Lasso and Double ML show negligible differences in terms of the sign, significance level, and magnitude of the impact of variables such as education expansion, education inequality, and parental dependency on upward mobility. This suggests that the estimates are robust. Among these three variables, parental dependency plays the most significant role (with an effect size of 0.10) in adjusting upward mobility, while education inequality plays the least significant role (with an effect size of 0.009). This conclusion indicates that while education expansion can promote social mobility, its effect is not substantial and can be partially offset by education inequality (about 1/6 of the effect), and may be entirely ineffective when parental dependency exists at a high level. These findings supplement the research results of Deirdre Bloome et al. [13] and Mitnik et al. [41] regarding how social inequalities undermine the effectiveness of education expansion policies in promoting upward mobility. Deirdre Bloome et al. [13] also argued that the expansion of higher education reduced persistence. However, this reduction in persistence was far from enough to offset the increase in persistence associated with growing educational inequality. New research conducted by Nobel laureate James J. Heckman and colleagues at the University of Chicago and the Rockwool Foundation in Denmark found that parents' and children's economic outcomes are much more tightly linked than previously believed, and therefore, current estimates of intergenerational mobility may be substantially overstated [42]. They developed new measures of economic welfare across the lifespan and found that the traditional analysis of family resources such as average income may have understated intergenerational dependence by 50% to 100% [42]. In agreement with James J. Heckman and his colleagues, my research has shown that parental influence overwhelmingly surpasses other factors when studying intergenerational mobility. Therefore, policies promoting social mobility should focus on changing perceptions in the tradition of 'like father, like son, like mother, like daughter'.

## 3.2 Is intergenerational mobility still seriously gender-biased?

Prior to the estimation process, an initial evaluation of gender bias in intergenerational mobility was conducted using the pre-modelled dataset sourced from the GDIM. Fig 6 presents a

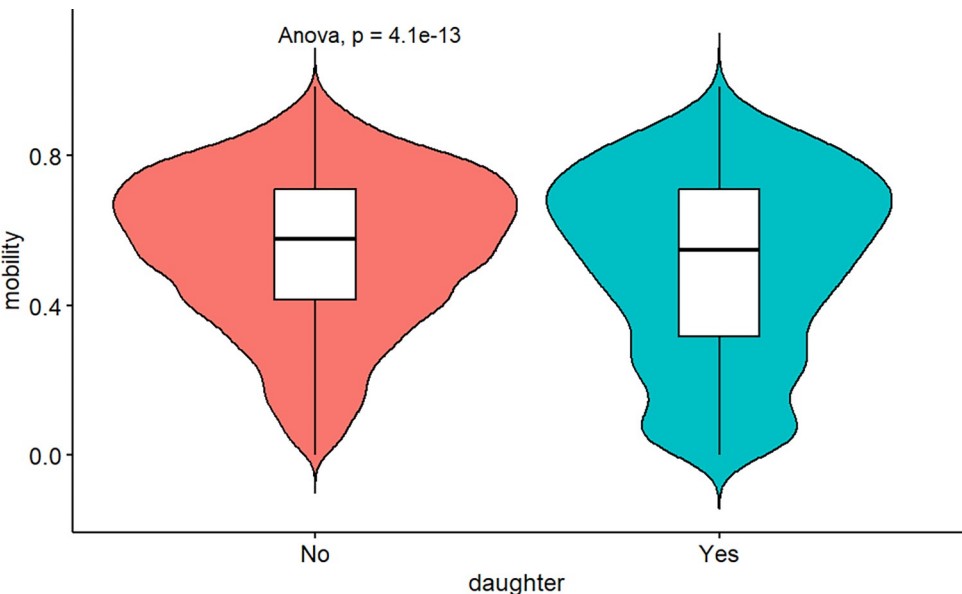

**Fig 6. Gender bias in upward mobility.** Notes: The sample size N = 6,725. The data for this estimation comes from the pre-modelled dataset. The method used for comparing means is ANOVA and p is its p-value.

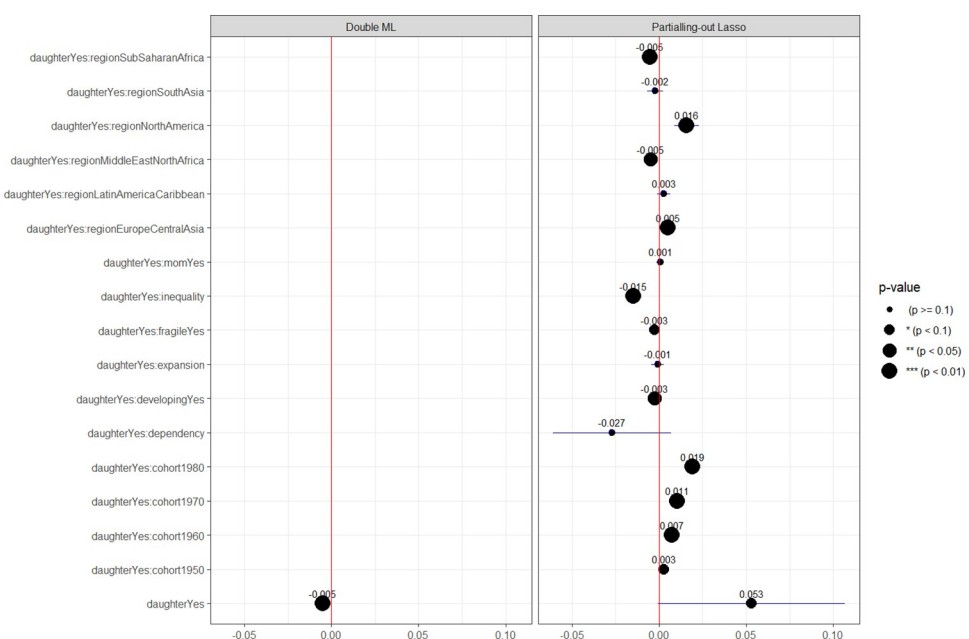

**Fig 7. Is intergenerational mobility still gender-biased?.** Notes: refer to the table format provided in S5 Appendix. The sample size N = 6725. The outcome variable is upward mobility.

comparison of the mean values of international mobility between daughter and non-daughter mobilities. The analysis employed the Analysis of Variance (ANOVA) method, with corresponding P-values reported for each case. The findings indicate that daughter mobility is statistically significantly lower (P<0.001), suggesting that intergenerational mobility may be gender-biased. Subsequently, the impact on mobility was estimated using Double ML to determine the overall effects. In addition, the Partialling-out Lasso technique was employed to measure the multi-session gender impacts on upward mobility through several factors: 'fragile' (indicating whether a country is classified as fragile or not), 'developing' (differentiating between developing and high-income countries), 'cohort' (in comparison to individuals born in the 1940s), and 'region' (compared to the reference region of East Asia & Pacific). The outcomes of Double ML and Partialling-out Lasso are visualized in Fig 7.

The Double ML estimates the overall gender effect on intergenerational mobility, and it reveals that daughters are still facing issues of inequality in social mobility, albeit to an insignificant degree. The overall gender effect size of 0.005 (p<0.005) is still too small to compare with the overall effect of parental dependency on social mobility (the size of 0.1 with p<0.001). This suggests that the gender effect is no longer a serious issue for social mobility, although it still exists to some extent. This result aligns with the trend of reducing gender inequality in social mobility that the World Bank has pointed out [30], or the research of Alberto Alesina et al. [31] that has not found clear evidence of gender inequality in mobility in the African region. This result may alleviate the excessive concerns of socio-economists about the issue of gender inequality in social mobility when a series of studies still point out (see the introductory section). With this research, I assert that although gender inequality still persists in mobility, it is far more important than the factors related to parental dependency. In addition, the Partialling-out Lasso estimation also shows that education inequality exacerbates gender inequality in mobility, with an effect size of 0.015 (p<0.005). However, when stripping away the characteristics of the context (country characteristics, regions, cohorts–see more in the data and methods), I have not found clear evidence of gender inequality at the family level (no

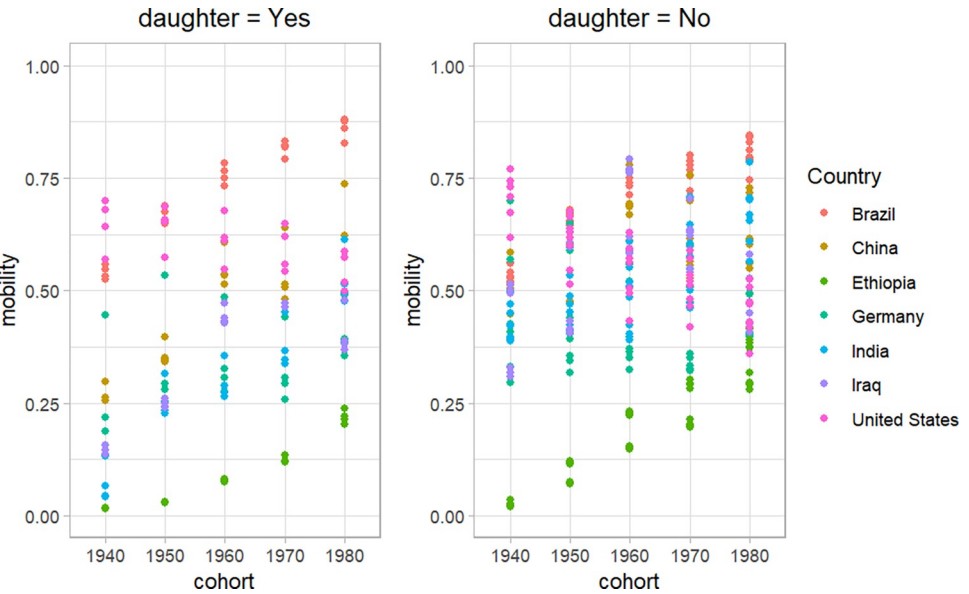

**Fig 8. The multidimensional gender bias in intergenerational mobility.** The data source for these charts is GDIM 2023. The chart on the left computes intergenerational mobility using the education levels of daughters. Conversely, the chart on the right uses the education levels of sons and all children in general to compute intergenerational mobility.

significant effect via parental dependency). This implies that the differential treatment of children is mainly influenced by social norms and cultures rather than natural tendencies. Therefore, this kind of gender effect can be observed in cultures that favour sons, such as India and China [21, 22].

Considering from a multidimensional perspective, Fig 8 also shows a trend of decreasing gender inequality in social mobility across generations. This is clearly demonstrated in Fig 7, where the interaction terms with cohort variables have positive values that increase over time. To clarify this conclusion, I extracted data from some representative countries for each region and plotted the changes in intergenerational mobility across the 1940s – 1980s generations. The graph showing daughters' mobility on the left clearly indicates an upward trend, while the graph on the right does not show this trend. Notably, daughters' mobility in India was very low in the 1940s but improved significantly by the 1980s generation. These findings align with the World Bank's conclusion on Global intergenerational mobility [30], which shows that inequality in this domain is gradually improving across generations. Fig 7 also shows a clear difference in gender bias in intergenerational mobility between regions, with the highest level of gender inequality observed in North America and the lowest in Africa. As shown in Fig 8, the US consistently has a high level of intergenerational mobility, while the lowest values of mobility are observed in Ethiopia. These findings contribute to the evidence that gender inequality is multidimensional and largely dependent on social norms and cultures, rather than inherent tendencies.

## 4. Conclusion

The escalating issue of global inequality has surfaced as a central concern in today's world, with intergenerational mobility serving as a crucial barometer of societal progress. The Great Gatsby Curve has illuminated the intricate dynamics between income inequality and the potential for individuals to climb the socioeconomic ladder. Education, an integral element of

this mobility, is acknowledged as a potent mechanism for disrupting the cycle of inequality. However, the impact of educational expansion on intergenerational mobility is tempered by societal inequality and parental influence. Despite the promise of educational expansion to catalyze upward mobility, its effectiveness is curtailed by persistent societal disparities and the enduring influence of familial factors. Moreover, the educational attainment of parents significantly shapes a child's trajectory, highlighting the powerful role of family circumstances in determining socioeconomic status. To fully comprehend the complex nature of intergenerational mobility, it is essential to examine gender inequality. Beyond societal and parental influences, gender biases within families can affect educational mobility. Yet, recent evidence suggests a closing gender gap, challenging traditional notions of gender inequality in intergenerational mobility.

The study presents an analytical framework that underscores the interconnectedness of educational inequality, parental influence, and gender effects in shaping upward mobility. By utilizing causal machine learning to untangle these complex relationships, the study enriches the existing literature. It leverages data from the Global Database on Intergenerational Mobility, covering 153 countries and cohorts born between the 1940s and 1980s. By applying Partialling-out Lasso and Double Machine Learning to address confounding issues, this research reveals that educational expansion can enhance intergenerational mobility to a certain extent. However, its effectiveness is partially offset by educational inequality and may be ineffective if high levels of parental influence persist. Furthermore, this study indicates that while gender inequality in intergenerational mobility persists, its extent has significantly reduced across generations. Compared to parental influence, gender effects are markedly less important. Therefore, there is an urgent need to reassess the roles of parental influence and gender bias in intergenerational mobility, especially considering the current underestimation of parental influence and overemphasis on gender bias.

## Supporting information

**S1 Appendix. The list of countries under this study.**
(DOCX)

**S2 Appendix. The pre-modelled dataset.**
(DOCX)

**S3 Appendix. The modelled dataset.**
(DOCX)

**S4 Appendix. Does education expansion work for promoting upward mobility?.**
(DOCX)

**S5 Appendix. Does gender bias exist in terms of upward mobility?.**
(DOCX)

## Acknowledgments

I am deeply grateful to Ciarán Nugent (the Nevin Economic Research Institute), Dr. Eoin Flaherty (Department of Sociology, Maynooth University), and Dr. Jan Skopek (Department of Sociology, Trinity College Dublin) for their invaluable comments and feedback on this research. Their expertise and guidance have been instrumental in shaping the study and enhancing its quality.

## Author Contributions

**Conceptualization:** Khanh Duong.

**Data curation:** Khanh Duong.

**Formal analysis:** Khanh Duong.

**Funding acquisition:** Khanh Duong.

**Investigation:** Khanh Duong.

**Methodology:** Khanh Duong.

**Project administration:** Khanh Duong.

**Resources:** Khanh Duong.

**Software:** Khanh Duong.

**Supervision:** Khanh Duong.

**Validation:** Khanh Duong.

**Visualization:** Khanh Duong.

**Writing – original draft:** Khanh Duong.

**Writing – review & editing:** Khanh Duong.

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
