## [Decision Letter · Decision Letter 0]

19 Dec 2023

PONE-D-23-22533Intergenerational Education Mobility: Does Education and Gender Equality Matter?PLOS ONE

Dear Dr. Duong,

Thank you for submitting your manuscript to PLOS ONE. After careful consideration, we feel that it has merit but does not fully meet PLOS ONE’s publication criteria as it currently stands. Therefore, we invite you to submit a revised version of the manuscript that addresses the points raised during the review process.

The reviewers have provided valuable feedback, highlighting the need for a more rigorous clarification and substantiation of your findings, particularly regarding the technical soundness and data support for your conclusions. On a positive note, they commend the appropriateness of your statistical analysis and data availability, as well as the clear use of standard English.

We look forward to receiving your revised manuscript.

Kind regards,

Rafi Amir-ud-Din

Academic Editor

PLOS ONE

Journal Requirements:

Reviewers' comments:

Reviewer's Responses to Questions

**Comments to the Author**

1. Is the manuscript technically sound, and do the data support the conclusions?

Reviewer #1: No

Reviewer #2: Yes

2. Has the statistical analysis been performed appropriately and rigorously? 

Reviewer #1: I Don't Know

Reviewer #2: Yes

3. Have the authors made all data underlying the findings in their manuscript fully available?

Reviewer #1: Yes

Reviewer #2: Yes

4. Is the manuscript presented in an intelligible fashion and written in standard English?

Reviewer #1: Yes

Reviewer #2: Yes

5. Review Comments to the Author

Reviewer #1: The authors have analyzed the relationship between education and gender inequality and intergenerational mobility using data from the Global Database on Intergenerational Mobility for 153 countries and cohorts born between the 1940s and 1980s. The paper specifically looks at two questions: first, whether education expansion policies mitigate the negative impacts of education inequality and parental influence on upward mobility; and second, whether gender inequality affect upward intergenerational mobility.

The authors have taken a painstakingly long exercise to answer questions which seem to have been thoroughly studied in literature and have been concretely established by multiple empirical studies. In other words, the research gaps this paper tries to fill are not clear. The authors also do not clarify how an analysis using the cohorts from 1940 to 1980 provides significant learnings for today’s policy and practice.

Some specific comments are below:

• The introduction is broken in two parts to justify the two research questions. Authors touch upon Great Gatsby Curve, importance of education, welfare-regime of the state, among the other predictors of inter-generational mobility, but the research question framed is limited only to education expansion policies given the inequality and parental dependency – even when authors cite other literature which investigate these relations. The literature cited do not justify the question framed.

• The second part of the introduction is supposedly about gender inequality. But line 68 – 76 mention other studies that study inequality.

• In line 91: Authors mention that there is a research gap regarding understanding gender inequalities that emanate from cultural and social norms. This is an incorrect assumption as there have been plethora of research now on understanding the multi-dimensionality of gender as well as its fluidity. Authors go on to cite evidence of multidimensional impact of gender but frame the research question without any mention of the same.

• Data and methods:

o The authors do not justify why they use education as an indicator for mobility and not income or wealth like many of the literature they cite.

o The indicator construction overall is unclear in the article.

o Education expansion does not necessarily reflect education policy, but authors use them interchangeably without justification.

o The last cohort of the Global Database on Intergenerational Mobility is 1980 whereas, authors include indicators of region and ‘fragility’ from a 2020 database. Other than the simple confirmation that the indicators of interest vary significantly by these country types, there is no concrete justification as to how a 2020 classification of countries apply to a 1980 cohort when there has been significant overhaul of policies both internally and globally.

o Authors talk about intergenerational mobility in terms of education without much justification. How does the probability of a child surpassing their parent's educational category, given that the parent does not have tertiary education reflect “expansion policies”? It can also reflect changes in country’s overall economic structure – and therefore, marginal propensity to consume.

o For both q1 and q2 models, there is no clarity on what the control variables are other than the indicators of interest. For any model on education, a number of predictors from both supply and demand side should be controlled for. Authors need to justify if no other controls are added.

o Authors note the contradictory results (Line 387-389: maternal education is strongly associated with immobility; but children are less dependent on their mother's education) but do not explain, interpret or discuss these findings at all.

o Authors make large assumptions based on country/region effects without clarifying or justifying these assumptions. If region specific results vary from overall results, are these results robust?

• The results dominate the write-up and discussion section does not go into interpretation of the results or try to place hypotheses from the counter-intuitive results. The methodology section also does not justify why the machine learning techniques were required or why a violin plot was needed to describe the data.

• Overall, the paper is poorly structured with no clear distinction between methodology, results, discussion, and conclusion. The results are also not placed in the larger context to clarify what this article is adding to a very old discussion.

Other than these, authors need to use consistent language, not use "females" when analyzing gender but use "women" and follow the structure of a scientific article.

The manuscript needs to be thoroughly read and restructured before it can be published.

Reviewer #2: This study used data from the Global Database on Intergenerational Mobility for 153 countries and those born between the 1940s and 1980s to examine the relationship between education and gender inequality on one hand, and intergenerational mobility on the other.

The aim was to:

1. Assess the effectiveness of education policies in promoting upward mobility

2. Investigate the direct and indirect effects of gender inequality on mobility outcomes.

The study found intricate interconnections among significant factors, such as educational expansion, educational inequality, and parental dependency, as they collectively shape mobility outcomes.

Even though what the study found is not so novel and we already know these, the author need to take time to reorganize the sections to make them more clearer and be more targeted in the recommendations. I have therefore suggested some actions below in this direction.

Analysis

1. With regards to your analysis, could you pull out the data on one or two countries in Sub-Saharan Africa and show how they contradict the East Asian and European countries? It will be good to se that analysis as shown for India, China, Denmark and the others.

Recommendations

2. Please unpick your recommendations and show them distinctively. For example:

- At which level of education would this investment you call for be made? Would grater educational investment be required at the basic, primary, pre-tertiary or at the tertiary levels? And please note, these must bring maximum impact.

3. The greater obstacles women face in achieving upward mobility should be addressed with which measures?

4. Is it only high educational levels that promote upward mobility? How about income levels of parents? And places such as India, how about extended family support systems that tend to enable people’s social mobility.

6. PLOS authors have the option to publish the peer review history of their article (what does this mean?). If published, this will include your full peer review and any attached files.

Reviewer #1: **Yes: **Ruchira Bhattacharya

Reviewer #2: No

---

## [Author Response · Author response to Decision Letter 0]

29 Dec 2023

Please find attached "Response to reviewers.docx"

---

## [Decision Letter · Decision Letter 1]

28 Mar 2024

What Really Matters for Global Intergenerational Mobility?

PONE-D-23-22533R1

Dear Dr. Duong,

We’re pleased to inform you that your manuscript has been judged scientifically suitable for publication and will be formally accepted for publication once it meets all outstanding technical requirements.

Kind regards,

Rafi Amir-ud-Din

Academic Editor

PLOS ONE

Additional Editor Comments (optional):

Reviewers' comments:

Reviewer's Responses to Questions

**Comments to the Author**

1. If the authors have adequately addressed your comments raised in a previous round of review and you feel that this manuscript is now acceptable for publication, you may indicate that here to bypass the “Comments to the Author” section, enter your conflict of interest statement in the “Confidential to Editor” section, and submit your "Accept" recommendation.

Reviewer #2: All comments have been addressed

Reviewer #3: (No Response)

2. Is the manuscript technically sound, and do the data support the conclusions?

Reviewer #2: Yes

Reviewer #3: Partly

3. Has the statistical analysis been performed appropriately and rigorously? 

Reviewer #2: Yes

Reviewer #3: Yes

4. Have the authors made all data underlying the findings in their manuscript fully available?

Reviewer #2: Yes

Reviewer #3: No

5. Is the manuscript presented in an intelligible fashion and written in standard English?

Reviewer #2: Yes

Reviewer #3: Yes

6. Review Comments to the Author

Reviewer #2: (No Response)

Reviewer #3: Dear Authors

This is a good work in terms of looking at the mobility and education relationship. I will recommend incorporating literature on integration of global markets to enhance the relevance of your study. Please refer to the following studies:

Gupta and Guidi (29012) Cointegration relationship and time varying co-movements among Indian and Asian developed stock markets

Guidi and Gupta (2013) Market efficiency in the ASEAN region: evidence from multivariate and cointegration tests

Beakert and Harvey (1995) Time-Varying World Market Integration

Since market integration has a bearing on inter generational mobility, it will benefit from providing a context here. However, I do realise that your study does not look at the mobility aacross nations but integration of markets provide important signal within the domestic context as well and the transmission of economic growth across countries.

Hope this helps

7. PLOS authors have the option to publish the peer review history of their article (what does this mean?). If published, this will include your full peer review and any attached files.

Reviewer #2: No

Reviewer #3: No

---

## [Editor Report · Acceptance letter]

5 Apr 2024

PONE-D-23-22533R1 

PLOS ONE

Dear Dr. Duong, 

I'm pleased to inform you that your manuscript has been deemed suitable for publication in PLOS ONE. Congratulations! Your manuscript is now being handed over to our production team.

Kind regards, 

on behalf of

Dr. Rafi Amir-ud-Din 

Academic Editor

PLOS ONE